# Molecular Mechanisms of Exercise and Healthspan

**DOI:** 10.3390/cells11050872

**Published:** 2022-03-03

**Authors:** Yuntian Guan, Zhen Yan

**Affiliations:** 1Department of Pharmacology, School of Medicine, University of Virginia, Charlottesville, VA 22903, USA; yg6ju@virginia.edu; 2Center for Skeletal Muscle Research at the Robert M. Berne Cardiovascular Research Center, School of Medicine, University of Virginia, Charlottesville, VA 22903, USA; 3Department of Medicine, School of Medicine, University of Virginia, Charlottesville, VA 22903, USA; 4Department of Molecular Physiology and Biological Biophysics, School of Medicine, University of Virginia, Charlottesville, VA 22903, USA

**Keywords:** healthspan, exercise, chronic diseases, adaptations

## Abstract

Healthspan is the period of our life without major debilitating diseases. In the modern world where unhealthy lifestyle choices and chronic diseases taper the healthspan, which lead to an enormous economic burden, finding ways to promote healthspan becomes a pressing goal of the scientific community. Exercise, one of humanity’s most ancient and effective lifestyle interventions, appears to be at the center of the solution since it can both treat and prevent the occurrence of many chronic diseases. Here, we will review the current evidence and opinions about regular exercise promoting healthspan through enhancing the functionality of our organ systems and preventing diseases.

## 1. Introduction

Two thousand and two hundred years have passed since the first Chinese emperor, Qin Shi Huang, ordered a nationwide hunt for the elixir for eternal life. Proven to be futile, his effort gave rise to the creation of a glorious terracotta army of 8000 that was buried alongside with him to embargo the pursuit for longevity [1]. Throughout the history of modern medicine, much like the emperor’s dream, scientists have never halted the search for ways to extend our lifespan. In the past 50 years, extensive research and development with tremendous amount of investment have led to an increase of lifespan in the U.S. by 10 years [2]. In 2019, there were over 703 million of the world population aged 65 or more; this number is projected to be over 1.5 billion by 2050 [3]. However, healthspan, the period of our life without major debilitating diseases, has not been prolonged [2]. Because aging remains the most important risk factor for nearly all chronic diseases [4,5], this creates a conundrum that the extended lifespan without improvement of healthspan leads to significant aging of the society and unsustainable economy.

Evidence and theories from research in the recent decades have shown that human life expectancy may have reached or come close to a limit set primarily by natural causes (chronic diseases) and genetics [6,7]. The concept of “healthy aging”, that is, to maximally expand the expectancy of healthy living before a person suffers from permanent aging-associated disabilities and chronic diseases, has emerged in the past two decades and gained significant popularity [8,9,10]. In the 21st century, there is a dire need of research on extending our healthspan [11,12,13]. Up to date, over 2100 active clinical trials in the United States are focusing on therapies that improve the quality of life (QOL) under the conditions of chronic diseases. Much like lifespan, healthspan can be influenced by numerous factors, namely genetics, environmental factors, social-economic status, lifestyle choices including dietary intake, physical activities, etc.

## 2. Exercise as Medicine to Promote Healthspan

The most ancient and potent “medicine” known to mankind that promotes healthspan is the engagement in organized, repeated and purposeful physical activities, or exercise training. The first documented exercise prescriptions by surgeons can be traced back to thousands of years ago in various ancient civilizations like the Ancient Greece and the Yellow River Civilization in China [14]. Over the recent decades, public health studies have shown indisputable evidence that high physical fitness is the most crucial factor for delaying all-cause mortality and the onset of chronic diseases, especially cardiovascular diseases, metabolic disorders and cancer [15,16,17]. Immense amount of research evidence has also demonstrated that long-term exercise training reshapes the molecular basis of multiple organ systems, including cardiorespiratory, musculoskeletal, neurological, endocrine and immune system [18,19,20]. For example, in a 12-year study in over 400,000 individuals, Wen et al. found the evidence that led to the conclusion that even 15 min a day or 90 min a week of moderate intensity exercise can reduce all-cause mortality [21].

Overall, exercise can be divided into two types: endurance and resistance exercise. Endurance exercise, often called aerobic exercise, is defined as exercise regimens that heavily rely on oxygen-utilizing energy metabolism, i.e., mitochondrial respiration, such as distance running, swimming and cycling. The main effects of endurance exercise include increase in mitochondrial content, capillary density, mass of slow-twitch oxidative myofibers and proportion of fast-twitch, oxidative myofibers in skeletal muscles along with enhanced cardiorespiratory function [22,23,24]. Resistance exercise, sometimes referred to as anaerobic exercise, relies primarily on anaerobic energy metabolism, namely creatine phosphate (CP) and anaerobic glycolysis for force production in a short period of time with less reliance on oxygen consumption. Typical resistance exercise regimens are heavy, low-rep weight training like bicep curls, leg extensions, sprinting and powerlifting. Adaptation of resistance exercise mainly include hypertrophy of the fast-twitch, glycolytic muscles, and usually does not or only moderately increase mitochondrial content [25,26]. Recent evidence suggests that although quantity of mitochondria might be unaltered, resistance exercise may still improve functionality of existing mitochondria, suggesting resistance exercise may also contribute to the regulation of mitochondrial quality [27].

Although the most direct executers of exercise are skeletal muscle and heart, regular exercise also leads to systemic changes in virtually all organ systems with superb, multifaceted benefits to health (Figure 1). In this review, we will summarize current opinions and evidence on how exercise training is important in promoting healthspan through enhancing the functionality of the cardiovascular system, skeletal muscle, nervous system and preventing the occurrence of cancer.

## 3. Cardiovascular System

### 3.1. Exercise Benefits in Promoting Cardiovascular Function

Cardiovascular diseases are the leading cause of all mortalities around the world, with sedentary lifestyle being one of the biggest risk factors [28,29]. The most direct clinical indicators of cardiorespiratory fitness are cardiac output and VO_2_max, both are major parameters that decline with aging, potentially a determining factor for healthspan [30]. Cardiac output is defined as the amount of oxygenated blood pumped by the left ventricle per minute and calculated as the product of stroke volume and heart rate. In a healthy individual, rigorous exercise may increase cardiac output by 4-fold; in trained elite athletes, this increase may be up to 6–8 fold [17]. VO_2_max is the maximal amount of oxygen that is utilized by the all the organ systems during exercise. Since the 1980s, it has been well categorized that VO_2_max increases with endurance exercise training regardless of age, sex or exercise mode, and is a critical measurement of risk factors of aging-related diseases [31]. A thorough analysis by Gries et al. showed that enhanced VO_2_max by exercise training during adulthood can even be kept up to the eighth decades in athletes [32].

One of the mechanisms proposed for exercise-mediated promotion of cardiovascular fitness is via physiological hypertrophy, i.e., an increase in heart muscle mass along with enhanced contractile function, as opposed to pathological hypertrophy along with pathological changes such as fibrosis that may lead to heart failure. Physiological hypertrophy occurs in response to increased wall stress and other signaling events during exercise training. A series of studies discovered that the insulin and insulin-like growth factor 1 (IGF1) pathways including insulin receptor substrate 1/2 (Akt1/2), IGF receptor and phosphoinositide 3-kinase (PI3K) are important in exercise-mediated physiological hypertrophy in mice [33,34,35,36,37]. In addition to hypertrophy, many other adaptations including contractile apparatus remodeling, mitochondrial remodeling, change in metabolism and angiogenesis along with altered gene expression also occur during exercise training [33,38,39]. Other pathways that are pertinent to physiological heart growth in response to exercise training remain to be fully explored. Therefore, future studies that focus on a more thorough investigation of the mechanisms behind exercise-induced physiological hypertrophy and enhanced cardiac function will provide new insights into healthspan benefits of exercise.

### 3.2. Exercise Benefits in Ameliorating Cardiovascular Diseases (CVD)

Endurance exercise has also been demonstrated to have both preventative and therapeutic effects on myocardial infarction [40,41,42,43,44], atherosclerosis [45,46], and hypertension [47,48,49], which are among the biggest risk factors that shorten our healthspan. Substantial amount of evidence have shown that long-term endurance exercise can delay aging-associated decline of both cardiac output and VO_2_max, contributing to disease prevention and promotion of quality of life [50,51,52,53,54]. In the past decade, the notion of “exercise as medicine” has gained tremendous popularity, which has prompted a number of clinical studies focusing on exercise regimens and methodologies to tackle cardiovascular challenges in patients [55,56,57,58]. Overall, it is recommended to prescribe moderate to low intensity (<60% maximal HR) endurance exercise at a frequency of 3–5 times per week to patients with cardiovascular diseases [55]. Significant effort has been committed in the past 50 years to unravel the molecular mechanisms behind exercise-mediated protection against CVD and promotion of recovery from cardiac injuries like ischemia-reperfusion (I/R); however, our understanding is still incomplete.

Heart failure is one of the top leading causes of death worldwide. Exercise is the best intervention for prevention and even treatment of heart failure without causing further cardiac injuries [59,60,61,62,63]. An early study in exercise intervention in human chronic heart failure patients in 1999 showed that one year of moderate intensity cycling (60% VO_2_max) led to better cardiac outcomes and quality of life improvements [64]. The mechanisms of exercise benefits on heart failure are multifaceted, without clear understanding of the exact molecular targets. Potential aspects that exercise may positively impact on preventing heart failure include enhanced energy metabolism [39,65], mitigated oxidative stress [66], improved mitochondrial function [64,67]; a clear understanding of the molecular mechanisms is yet to be investigated.

Atherosclerosis is a prevalent, chronic vascular disease that affects millions of people worldwide. The pathology of atherosclerosis is complicated and multifaceted, with inflammatory factors and endothelial activation being early contributors; eventually, arterial wall lesions form, and cholesterol-rich lipids form blockades in the vessels, causing detrimental damage. Exercise has been discovered to (1) enhance endothelial function, and (2) exhibit anti-inflammatory and antioxidant effects as ways to help prevent the onset of atherosclerosis [68,69,70,71,72,73,74,75,76].

Overall, molecular mechanisms underlying exercise benefits in CVD is multi-faceted and incompletely understood. First and foremost, physiological stress signaling induced by exercise may be key. The most studied stress signaling kinase in exercise is 5′ AMP-activated kinase (AMPK) since it is activated during energetic stress, i.e., increased AMP/ADP concentration [77]. Studies over the years have found it to be a critical signaling molecule in CVD like ischemia/reperfusion (I/R), fibrosis and vascular dysfunction [78,79,80,81,82]. AMPK is a heterotrimeric complex with a catalytic α domain, a non-catalytic β domain and a regulatory, AMP/APD-binding γ domain. Each subunit has different isoforms that are encoded by separate genes, which determines AMPK expression of different tissues (α1, α2, β1, β2, γ1, γ2, and γ3). In heart, α2-, β2- and γ1-subunit are most commonly expressed [83]. Acute exercise causes energetic stress and signaling events that are sensed by AMPK and activates AMPK, leading to a series of phosphorylation cascades to regulate downstream signaling. Recent evidence also suggested that subcellular localization of AMPK may also be of importance, e.g., exercise may specifically activate mitochondria-associated AMPK [84]. Future studies on subcellular AMPK pools will improve the understanding in how AMPK regulates exercise benefits.

Mitochondria may also be a key player in exercise-mediated protection from CVD due to its central role in CVD progression [85,86]. Exercise training can potentially both improve mitochondrial content (biogenesis) and dynamics (fission, fusion and mitophagy). It has been well-established that increases in transcription and abundance of peroxisome proliferator-activated receptor co-activator (PGC-1α), a master transcriptional co-activator for mitochondrial and oxidative metabolism, is responsible for increased mitochondrial biogenesis after exercise [87,88,89]. In addition, recent evidence suggests that the improved quality of mitochondria, alongside with increase in quantity, may also be a key factor in protecting the heart from CVD [67,90,91,92]. Mitochondrial quality is usually measured with O_2_ consumption, membrane integrity and morphology, which can all be enhanced by regular exercise [93,94,95,96,97,98]. A 2019 study found that mitophagy, more than macroautophagy, is critical in maintaining cardiac function in diabetic cardiomyopathy [99]. Although the necessity of exercise-induced mitophagy is not explored in this model, it has been suspected that, in other cardiac injury models, exercise may help maintain cardiac function through upregulating mitophagy [100,101,102].

Last but not least, skeletal muscle-derived humoral factors by endurance exercise may provide protection against pathological development of heart failure under disease conditions. For example, extracellular superoxide dismutase (EcSOD) is a superoxide scavenger that is upregulated in skeletal muscle upon exercise and travels to the peripheral tissues/organs, including the heart through the circulation, which has been shown to prevent diabetic cardiomyopathy [103,104].

## 4. Skeletal Muscle

### 4.1. Exercise Benefits in Skeletal Muscle Mass & Strength

Skeletal muscle quality, comprised of muscle mass (the number and size of muscle fibers) and strength (force production and contractility of muscle fibers), is one of the most important factors for quality of life as it is crucial for mobility, balance, motor coordination. Importantly, loss of muscle mass and strength both contribute to decreased healthspan [105,106]. Aging-associated loss of skeletal muscle mass (sarcopenia) and frailty are among the most prevalent causes of morbidity in aged population, affecting over 10% of the population over 60 years of age [107,108,109]. However, loss of muscle mass can be significantly delayed or even prevented by regular exercise, lengthening healthspan [110,111]. Clinical studies have found that aging may account for ~20–25% loss of muscle cross-sectional area compared to that of young individuals [112,113]. Importantly, anaerobic exercise is able to significantly delay the aging-associated sarcopenia as measured by muscle cross-sectional area or muscle mass [113,114,115]. Some evidence also suggests that endurance exercise can also improve sarcopenic conditions [116,117]

The mechanisms underlying exercise-mediated promotion and conservation of muscle mass is poorly understood. Several targets have been identified as important players in regulating muscle mass. Myostatin is a transforming growth factor-β (TGF-β) superfamily member that inversely regulates muscle growth and is upregulated in patients with sarcopenia [118]. Both endurance exercise and resistance exercise decrease muscle and plasma levels of myostatin, which may contribute to the mitigation of muscle wasting [119,120,121,122]. A combination of siRNA-mediated knockdown of myostatin and endurance exercise training has been shown to promote skeletal muscle hypertrophy with activation of resident stem cells [123]. Studies have also shown that protein synthesis via anabolic signaling, such as the Akt-mTOR pathway, is also associated with improvement in muscle mass by exercise training [124,125]. Interestingly, both rapamycin-sensitive and -insensitive mTOR seem to be important for resistance exercise-induced muscle protein synthesis [126,127]. In addition, proteolytic ubiquitin ligases, like MuRF-1 and Atrogin-1, which contribute to muscle wasting, can be reduced by exercise training [128]. Altogether, the mechanisms exercise benefits are clearly multifaceted. Therefore, future studies with genetic animal models are needed to improve our understanding.

Muscle strength, the ability of maximal force production per unit of muscle mass, is also important for quality of life. This is measured by the amplitude and velocity of muscle contraction, which are usually impaired as we age [129]. Resistance exercise improves muscle function also through improving muscle strength. Healthy young men can have 30–42% increase of fiber peak power after a 12-week resistance exercise intervention [130]. Endurance exercise has also been shown to improve the contractile profile of muscle fiber [131,132,133]. Interestingly, women in their seventies with lifelong aerobic exercise training had increased strength in type I fibers and increased contractile velocity in type IIa fibers without increase in fiber size or mass compared to sedentary counterparts [134]. A possible mechanism is the enhanced Ca^2+^ sensitivity, defined as the force produced when the fiber is exposed to a given submaximal Ca^2+^ concentration. Although no change was observed after sprint training for type I, IIa, or IIa/IIx fibers [135], and no difference was observed between master runners and sedentary individuals for type I and IIa fibers [136], the effects of resistance training on Ca^2+^ sensitivity has been observed in type I fibers in old women [137].

The clinically relevant assessment of skeletal muscle function is exercise capacity, often measured by treadmill running, ergometer bike or 6-min walk. Importantly, exercise capacity is inversely correlated with all all-cause mortality [138], and numerous randomized clinical trials showed that exercise interventions improve exercise capacity in various health and disease populations [139,140,141]. Underlying the improved exercise capacity upon exercise training, particularly endurance exercise training, are a variety of physiological and biochemical adaptations in skeletal muscle, including mitochondrial biogenesis, angiogenesis, and fiber type transformation. These adaptive changes are the basis for the improvement of physical performance and other health benefits. Specifically, fiber type transformation induced by endurance exercise training in the direction of type IIb/IId/x to IIa fibers appear to be caused by activation of the calcineurin-nuclear factor of activated T-cells (NFAT) pathway [142,143,144]. Exercise training also induces adaptations closed related to regulation of energetic homeostasis. Exercise training, particularly endurance exercise, induced mitochondrial biogenesis in skeletal muscle, which is caused by induced expression/activity of transcriptional co-activator, peroxisome proliferator-activated receptor γ co-activator 1α (PGC-1α), coordinating the transcription of the mitochondrial and nuclear genomes for new mitochondrial biogenesis [88,145,146,147]. Finally, improved exercise capacity by endurance exercise training is also associated with angiogenesis, an expansion of the capillary network from preexisting capillaries in recruited skeletal muscles to improve gas and nutrient delivery. PGC-1α has emerged as a key regulator of angiogenesis in skeletal muscle in a hypoxia-inducible factor (HIF)-independent manner where PGC-1α coactivates the orphan nuclear receptor estrogen-related receptor-α (ERRα) [148]. Whole body *Pgc-1α* gene deletion led to reduced VEGF protein expression and blunted response to acute and chronic exercise training [149]. Importantly, muscle-specific deletion of the Pgc-1α gene led to significant attenuation of contractile activity-induced VEGF expression and exercise-induced angiogenesis but not fiber type transformation [147,150]. Muscle-specific deletion of the *Vegfa* gene led to significantly reduced capillary density and exercise training-induced angiogenesis in skeletal muscle [151,152].

In summary, a sophisticated signaling-transcription network mediates exercise-induced skeletal muscle adaptations, leading to improved mass, strength and endurance capacity. These improved contractile functions profoundly promote healthspan. Continued research efforts will elucidate the highly coordinated remodeling processes in skeletal muscle and unveil further the importance of skeletal muscle health in healthspan.

### 4.2. Exercise Benefits in Skeletal Muscle Metabolism

Metabolic diseases like insulin resistance and obesity are one of the major roadblocks to healthspan. Skeletal muscle, an organ that accounts for ~40% of body weight, is responsible for the majority of postprandial glucose uptake [153]. Fortunately, skeletal muscle is also one of the biggest beneficiaries of exercise intervention, making many metabolic diseases, such as type II diabetes, avoidable or delayable by lifestyle intervention [154]. For long, it has been shown that muscle contraction stimulates increased insulin sensitivity in muscle [155,156,157]. Long-term studies also found exercise training in a prolonged period result in improved insulin sensitivity [158,159,160,161,162]. Molecularly, it was established that muscle contractions during exercise lead to stimulation of translocation of glucose transporter 4 (GLUT4) vesicles to plasma membrane, significantly elevating glucose uptake [163,164], which is controlled via exercise-induced AMPK phosphorylation of TBC1D1 [165,166,167]. In addition, mitochondrial content and quality is also upregulated in response to exercise, contributing to more efficient energy production during prolonged exercise [87,168,169,170,171]. More recently, mitochondrial dynamics that control mitochondrial quality, i.e., fission, fusion and mitophagy, has been brought up as another important aspect of exercise benefits in skeletal muscle metabolism [172,173,174]. Interestingly, nearly all of the mitochondria-related mechanisms of exercise seem to require the signaling of AMPK, the energetic sensor activated by exercise [175]. Recent evidence by our lab have discovered that a unique pool of AMPK is localized to mitochondria; others have also suggested that AMPK may be localized to other organelles in regulating key pathways [84,176]. Future studies in organelle-specific AMPK may greatly contribute to the understanding of how exercise induces metabolic improvements in skeletal muscle.

### 4.3. Exercise-Induced Muscle-Derived Antioxidant

Clinical and animal studies in the past decade have found that oxidative stress caused by uncontrolled overproduction of reactive oxygen species (ROS) is a major problem underlying many conditions like aging, diabetes and cardiometabolic diseases, making it one of the biggest impediments in increasing healthspan [177,178]. Exercise-induced antioxidant actions have then been proposed to promote health in this regard, however its mechanisms still remain incompletely understood [179,180,181].

The idea of exercise-mediated promotion of antioxidant system is different from the one of pharmacological supplementation. Exercise causes a “physiological” increase in oxidative stress in skeletal muscle, which then turns on physiological pathways that increase enzymatic responses to counteract oxidative damage, thus benefiting other tissue/organs as well [182,183]. Recently, studies on superoxide dismutases (SODs), a family of enzymes that neutralize superoxide anions (O_2_^−^) as the first line of defense against oxidative stress, have proposed that these enzymes might be important in exercise-mediated benefits through enhancing antioxidant system [184]. In particular, extracellular superoxide dismutase (EcSOD) is the only known antioxidant that has a capacity of scavenging ROS on cell surface and extracellular matrix, gaining much attention [185,186]. A study from our lab in 2015 showed that EcSOD is primarily expressed in skeletal muscle and can be upregulated by exercise training, which then travels through circulation and accumulates in primarily heart and lung, exerting antioxidant effects in the condition of diabetic cardiomyopathy [187]. A follow-up study in 2017 further demonstrated that skeletal muscle derived EcSOD, through genetic overexpression and serum transfusion, generates protective effects through inhibition of endothelial activation to protect mice in a model of multi-organ dysfunction [187]. Altogether, this evidence showed that cross talks between muscle and other organs in terms of strengthening whole-body antioxidant capacity may partly explain exercise benefits in prolonging healthspan. Future studies are also warranted to expand current knowledge in the detailed mechanisms of how EcSOD and other myokines may achieve this protection.

## 5. Adipose Tissue

Obesity, measured by body mass index in the general public, is a complex disease involving an excessive amount of body fat. Obesity has risen to a pandemic proportion in the U.S. and has become a major global health problem [188]. It is one of the most serious metabolic syndromes that correlate with many other diseases, like cardiovascular diseases and diabetes, becoming a huge burden on healthspan. To date, the most efficient countermeasure of obesity is lifestyle managements, including healthy diet and physical exercise [189,190,191]. Mechanistically, excessive adiposity leads to decreased insulin sensitivity in skeletal muscle, potentially through lipotoxicity to mitochondria [192,193,194]. Excessive adiposity is also detrimental due to the production and secretion of pro-inflammatory and pro-oxidant factors [195,196,197,198]. The exact mechanisms underlying these pathways are still not fully understood. On the other hand, exercise is known to induce fatty acid oxidation in order to meet the energy demand [199,200], and long-term exercise training appears to enhance the capacity of muscle to uptake fatty acids as well as fatty acid oxidation [201,202,203,204]. Future studies should investigate the molecular targets of exercise benefit in adipose tissue in these regards.

Adipose tissue is also a metabolic organ in mammals, consisting of primarily white adipose tissue (WAT) and brown adipose tissue (BAT). WAT stores energy in the form of triglycerides, whereas BAT is responsible for shivering thermogenesis, a process that generates heat through uncoupling mitochondria when activated [205]. It was later discovered that adipose tissue browning, a conversion of WAT to BAT, may positively contribute to metabolism and negatively correlated with aging, making it a potential important aspect of prolonging healthspan [206,207]. Exercise-induced browning of white adipose tissue is well documented [208,209]. However, its mechanism still remains elusive. A study in 2019 showed that deletion of the fibronectin type III domain containing 5 (Fndc5) gene in mice which produces irisin, a peptide that has been shown to induce WAT browning, leads to significantly less exercise-induced metabolic benefits [210]. Another study reported that interleukin 6 (IL-6) is required for both baseline expression and exercise-induced upregulation of uncoupling protein 1 (UCP1), the main molecular mediator of thermogenesis in BAT [211]. Therefore, future studies that elucidates the mechanisms behind exercise-induced browning of WAT is still warranted to generate a clear understanding of how exercise-mediated muscle-adipose tissue crosstalk may prolong healthspan.

## 6. Liver

Non-alcoholic fatty liver disease (NAFLD) is defined as liver disease caused by accumulation of fatty acids and not by alcohol use. NAFLD is a globally prevalent disease that includes a range of conditions ranging from hepatic steatosis to non-alcoholic steatohepatitis (NASH), fibrosis and cirrhosis; it may eventually progress into liver failure, becoming a major impediment that reduces healthspan [212,213]. NAFLD is largely considered a hepatic manifestation of metabolic syndrome. Therefore, one would reasonably argue that metabolic exercise benefits would lead to improved or delayed onset of NAFLD. Indeed, two independent randomized controlled clinical trials have confirmed that exercise training, either aerobic or resistance training, could reduce hepatic fat in NAFLD patients [214,215].

Since exercise effects are extremely multifaceted and NAFLD may benefit from “secondary” effects such as weight loss or improved insulin sensitivity, it is difficult to narrow down to specific pathways regarding the molecular mechanisms of exercise benefits in NAFLD. On top of that, some inconsistencies between animal phenotypes of NAFLD models and human patients also create difficulties in hepatic factors that may mediate exercise benefits [216]. Nevertheless, several studies in rodent models suggest that improved insulin resistance, reduced hyperlipidemia and reduced hepatic inflammation are potential mechanisms of exercise benefits [217,218,219]. One study in 2014 suggested that exercise was able to restore impaired hepatic mitochondrial respiration in a mouse NASH model, raising a possibility that exercise benefits may be mediated by mitochondrial regulation [220]. Altogether, future mechanistic studies are needed in order to formulate a more complete understanding of the molecular actions in exercise benefits in NAFLD.

Fibroblast growth factor 21 (FGF21) is a hormone that is primarily released in liver and muscle, and can be significantly upregulated during stress conditions, like exercise, which then can localize to different tissue/organ through circulation [221,222,223,224]. A study in 2016 showed that FGF21 is required in improving glucose tolerance and hepatic triglyceride in rats after voluntary wheel running, demonstrating the potentials of FGF21 to be an important mediator of exercise benefits [225]. Another study showed that FGF21 is required in hepatic mitochondrial function. However, the functional role of FGF21 appears to be paradoxical; its regulators and exact mechanisms of action are not clearly understood. For example, a study in 2019 showed that FGF21^−/−^ mice are protected from fasting-induced muscle atrophy potentially through regulating Bnip3-mediated mitophagy [226]. Therefore, it is critical for future studies to focus on the up-/downstream regulators of FGF21 to fully understand how FGF21 mediates exercise benefits in healthspan.

## 7. Central Nervous System

### 7.1. Exercise-Mediated Benefits on Neurological Health

Neurological disorders have become a tremendous burden in the United States and the world, being projected to cost over $16 trillion USD by 2030 [227,228,229]. Common neurological disorders that contribute to loss of quality of life and are worsened by aging include Alzheimer’s disease, dementia, Parkinson’s disease, multiple sclerosis, epilepsy, etc. In the past decades, substantial evidence suggest that regular exercise is clearly the most potent method in both mitigating and preventing cognitive decline [230,231,232,233,234,235,236,237,238], Parkinson’s disease [238,239,240], multiple sclerosis [241,242,243,244,245,246,247,248,249], and depression [250,251,252]. In all age groups, regular exercise has been shown to enhance central nervous system functions including but not limited to cognitive function, coordination, visuospatial memory, and learning abilities [253,254,255,256,257]. A recent study by Hatch et al. found that even a 30-min high intensity intermittent exercise is sufficient to enhance cognitive function in young adults [255]. Another recent clinical trial by Carta et al. demonstrated that a 12-week moderate intensity exercise with a combination of endurance, resistance exercise and balancing activities significantly improved cognitive function in healthy elderly people [256]. Of note, clinical studies and meta-analyses did not find that vigorous exercise regime is superior in enhancing brain function compared to moderate exercise, suggesting that an “ideal dose” of exercise may exist, and other factors need to be considered in determining the best exercise program for different populations [258,259,260]. This notion encouraged numerous studies that evaluated different modes of exercise in enhancing brain function [261,262], such as high intensity interval training (HIIT) [257], coordinative exercise [263,264], mind-body exercise like Tai Chi [265,266], etc. Overall, regular exercise remains to be the most prominent life-style intervention in improving healthspan with regard to the mental/neurological function.

### 7.2. Mechanisms of Exercise Benefits on Central Nervous System

#### 7.2.1. Neuroplasticity

Neuroplasticity describes the process and capacity for neuronal network to undergo structural and functional changes as it adapts to behavioral stimulations, like exercise. In both rodent and human exercise models, studies have shown that endurance exercise training increases brain volume of different regions such as the hippocampus [267,268]. Exercise training also enhances dendrite length and complexity in many areas of the brain, including but not limited to the hippocampus, basolateral amygdala area, medial prefrontal cortex, and nitrergic neurons in the cerebral cortex [269,270,271,272]. All these morphological changes in brain volume and dendrites are thought to contribute to the exercise-induced improvements in motor skill, memory and cognitive functions, as well as protection from neurodegenerative diseases [273,274,275].

Exercise-mediated neuroplasticity enhancement is often associated neuronal functional changes, which are multi-faceted [276]. Exercise may improve long-term potentiation, a process that describes the long-term strengthening of neuronal communication/signal transmission due to persistent, patterned stimulation, in the hippocampal area in both young and aged rodents, and this adaptation appears to be intensity- and duration-dependent [277,278,279,280,281,282]. A study in 2015 showed that 12 days of voluntary wheel running also increased hippocampal astrocytic markers and altered astrocyte morphology, suggesting glial function may also play a role in enhanced hippocampal plasticity by exercise [283]. Altogether, this evidence clearly demonstrated that neuroplasticity is an important aspect of exercise benefits in healthspan. Future investigations on molecular mediators in these pathways are essential.

Brain-derived neurotrophic factor (BDNF) was among the first molecules discovered to be important in exercise-mediated enhancement in neuroplasticity [284,285]. In an early study in rats, expression of BDNF and its receptor tropomyosin receptor kinase B (TrkB) both increased dramatically after voluntary wheel running exercise [286], which is confirmed by later studies with different exercise modes and in different ages [287,288]. Conversely, pharmacological inhibition of TrkB blocked the exercise-induced increase in neuroplasticity markers [286]. Indeed, human studies also reported strong correlation between exercise-induced BDNF and improved cognitive function, involving the hippocampal area [289,290]. However, the mechanisms on the regulation of BDNF and its downstream targets by exercise are elusive. An early study showed that blocking hippocampal insulin-like growth factor (IGF-1) receptor reversed the induction of exercise-stimulated BDNF expression, suggesting the importance of IGF-1 [291,292]. Blocking of IGF-1 receptor also blunted exercise-induced synapsin I expression, calcium/calmodulin protein kinase II (CaMKII) and mitogen-activated protein kinase II (MAPKII) phosphorylation in the hippocampus [291]. Interestingly, recent evidence suggests that aerobic exercise and resistance exercise, although both improves cognitive function may act through different downstream molecular pathways [293,294]. In conclusion, future studies in molecular pathways involving BDNF, IGF-1 and their downstream cascades are crucial in understanding how neuroplasticity plays a pivotal role in exercise benefits in brain health.

#### 7.2.2. Angiogenesis

On the cellular level, endurance exercise has been shown to reshape cellular pathways that promotes neurogenesis, angiogenesis, neuronal plasticity, and vasculature function in the brain [295,296,297]. It is now established that VO_2_max is a predictor of vascular function in the brain, which is closely associated with maintenance of cognitive abilities during aging [298]. Ainslie et al. found that blood flow velocity in the middle cerebral artery (MCAv) in the endurance exercise trained individuals is ~17% greater than the sedentary counterparts in almost all age groups (18–79), even though both groups still suffer from the aging-induced loss of MCAv [295]. Akazawa et al. showed that a 12-week cycling intervention was able to significantly enhance the cerebral microvascular tone in older adults compare with the sedentary control group [299].

#### 7.2.3. Neurogenesis

Long-term endurance exercise can significantly delay the loss of neuronal volume in the hippocampus area, which underlies some cognitive disorders associated with aging, such as dementia and Alzheimer’s disease [267]. Several molecules with neurogenesis-stimulating potentials have been shown to be important in exercise-induced neurogenesis in the hippocampus area, including BDNF, insulin-like growth factor-1 (IGF-1), and vascular endothelial growth factor (VEGF) [285,300,301,302]. Importantly, studies have suggested that exercise-induced BDNF is significant in promoting neurogenesis in the elderly population, emphasizing the importance of this mechanism in neuronal healthspan [303]. Cathepsin B (CTSB) is a newly identified myokine from skeletal muscle that is upregulated by aerobic exercise and crosses blood-brain barrier to potentially regulate brain biochemistry, although its function and whether it directly acts on promoting neurogenesis are not completely understood [304].

Resistance exercise is also effective in preserving cognitive function although the evidence is relatively limited, and its mechanism(s) is yet to be investigated [305]. One study showed that a 24-week of either moderate- or high-intensity resistance exercise was able to improve cognitive function in older adults [306]; a similar study showed that resistance exercise at the rate of once or twice a week is effective in enhancing cognitive function among older women [307]. In conclusion, the molecular and cellular mechanism(s) behind exercise adaptations that result in enhanced neurological function is still elusive. Interestingly, a recent meta-analysis of clinical studies showed that an increase in BDNF is more significant when training regimes incorporate more resistance training than moderate-intensity endurance exercise in older adults [308]. More mechanistic studies in the future are needed to improve our understanding of how regular exercise enhances neurological health hence our healthspan.

## 8. Cancer Prevention

Cancer claims more than 600,000 lives each year in the US alone [309]. For long, it has been known that regular exercise is very effective in preventing many if not all kinds of cancers [310]. Cancer cachexia, weight loss, and cognitive decline are among the most detrimental comorbidities associated with cancer that greatly compromise healthspan. Cancer begins with mutations of tumor suppression genes that control normal cell growth, which then causes uncontrollably growing, transforming regular cells into tumor cells. If enabled by matured tumor microenvironment, tumor cells may eventually develop into full-blown cancer [311]. Extensive clinical trials and meta-analyses reveal that the level of physical activity or regular exercise positively correlates with lower risks of many types of cancer, including at least colon, breast, kidney, endometrial, bladder, esophagus, stomach and lung cancers [312,313,314,315,316]. Research on how exercise training can be used as a primary or secondary treatment and prevention of cancers has been proposed and will have huge societal impact [317,318].

The prominent hypotheses about the molecular mechanism(s) underlying the anti-cancer effects of regular exercise are proposed around exercise-mediated tuning of the immune system, which can be circumvented by tumor cells via microenvironment [319]. Given its known beneficial effects to the immune system, regular exercise may help to inhibit the development of tumor microenvironment by improving the innate and adaptive immune system [317]. A recent study by Pedersen et al. showed that voluntary wheel running induces infiltration of natural killer (NK) cells in tumor tissues in a mouse model of subcutaneous melanoma, which may contribute to the inhibition of tumor growth [320]. Another potential mechanism is that exercise training reduces overall cellular oxidative stress, which has been linked to oncogenesis [321,322]. We have previously shown that endurance exercise results in upregulation of extracellular superoxide dismutase (EcSOD) in skeletal muscle, which circulates to the heart, lung and other peripheral organs and protects them from oxidative damage [104,187]. In conclusion, there is a strong need to unravel the mechanism(s) underlying exercise-mediated anti-cancer effects in order to understand the broader impacts of exercise on prolonging our healthspan.

## 9. Conclusions

In summary, exercise training remains the most potent “medicine” that preserves quality of life and expands healthspan. The molecular understanding of exercise impacts in different organ systems reinstates that exercise is the most powerful lifestyle intervention against chronic diseases. While human lifespan seems to approach its limit, great potentials lie in promoting physical activities among any given communities to improve the healthspan and possibly lifespan as well.

## Figures and Tables

**Figure 1 cells-11-00872-f001:**
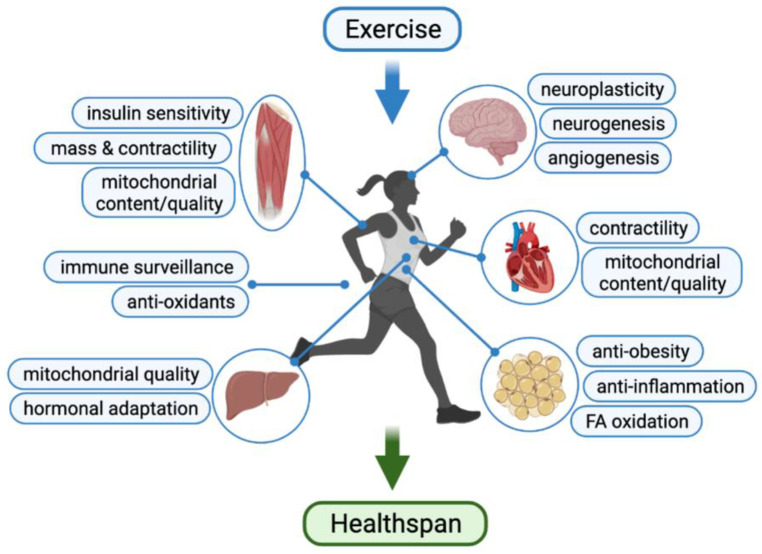
Illustration of exercise benefits in prolonging healthspan.

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
