# Peer review of "Molecular Mechanisms of Exercise and Healthspan"

_cells, 2022, doi:10.3390/cells11050872_

Round 1

Reviewer 1 Report

Dear Authors, 

Your manuscript was interesting, well written and an easy read.  However, it was written from a very broad perspective and appropriate for a lay audience rather than a scientific audience. Moreover, it really did not touch on molecular mechanisms. Most of the paper was related to the basic understanding that exercise is beneficial.  The paper only superficially touched on mechanisms by briefly highlighting myokines, AMPK, NKs.  And then oddly, added in specific sentences related the author’s research (Lines 122-128, 157-159, 231-233).  In some sections, no molecular mechanisms were even given (line 205-207).

Essentially, this manuscript did not comprehensively review the molecular mechanisms of exercise and health span but rather, gave a very brief review of the beneficial effects of exercise on striated muscle, neurological health, and cancer. 

Suggestions for the Authors:

Identify your target audience.

Adjust title and purpose.

Be more comprehensive.

Do not oversimply too much (e.g., although classically accepted, newer studies do suggest resistance training can influence mitochondria - PMC4478283.)

Be less obvious when trying to highlight your own work.

Figures would be helpful.

Author Response

Reviewer 1:
Your manuscript was interesting, well written and an easy read.  However, it was written from a very broad perspective and appropriate for a lay audience rather than a scientific audience. Moreover, it really did not touch on molecular mechanisms. Most of the paper was related to the basic understanding that exercise is beneficial.  The paper only superficially touched on mechanisms by briefly highlighting myokines, AMPK, NKs.  And then oddly, added in specific sentences related the author’s research (Lines 122-128, 157-159, 231-233).  In some sections, no molecular mechanisms were even given (line 205-207).
Essentially, this manuscript did not comprehensively review the molecular mechanisms of exercise and health span but rather, gave a very brief review of the beneficial effects of exercise on striated muscle, neurological health, and cancer.
Suggestions for the Authors:
Identify your target audience.
Adjust title and purpose.
Be more comprehensive.
Do not oversimply too much (e.g., although classically accepted, newer studies do suggest resistance training can influence mitochondria - PMC4478283.)
Be less obvious when trying to highlight your own work.
Figures would be helpful.
Response:
We thank the reviewer for the critical yet constructive comments and suggestions. We agree that the previous version of the review on mechanisms of exercise benefits on healthspan was superficial and brief. Therefore, we made the following major revisions:
1. Added to each section more detailed discussions of mechanisms of exercise benefits in healthspan and potential future directions (line 96-112, 138-180, 205-226).
2. Expanded the review and discussion of exercise mechanisms on neurological health (line 321-357).
3. Added sections discussing healthspan benefits of liver, adipose tissue, and antioxidant effects (multi-organ) to cover broader areas of healthspan.
We hope that the newer version covers more molecular mechanisms in exercise benefits in healthspan and provides useful information to the audience.
• We also edited the texts so that when works from our lab are mentioned, they are in accordance with the discussion of the topics (line 223, line 406-409).
• Added a figure to summarize the topics in the review.

Reviewer 2 Report

The present manuscript reviewed existing evidence about the role of regular exercise in promoting healthspan. The review structure is straightforward, comprehensive and fluent. The manuscript is overall well-argued and well-written. Pay attention to third-person singular forms of verbs (eg, line 20) and typing errors, eg. lines 131, 138). References and citations claim the topic.

As an adjunct, authors should strengthen the paragraph about the “Excercise-mediated promotion of cardiovascular function” by depeening some pathways and molecular mechanisms contributing to the cardiovascular benefits of exercise, as done for the discussion of the exercise benefits in skeletal and striated muscle, neurological disorders and cancer.

Author Response

Reviewer 2:
The present manuscript reviewed existing evidence about the role of regular exercise in promoting healthspan. The review structure is straightforward, comprehensive and fluent. The manuscript is overall well-argued and well-written. Pay attention to third-person singular forms of verbs (eg, line 20) and typing errors, eg. lines 131, 138). References and citations claim the topic.
As an adjunct, authors should strengthen the paragraph about the “Exercise-mediated promotion of cardiovascular function” by depeening some pathways and molecular mechanisms contributing to the cardiovascular benefits of exercise, as done for the discussion of the exercise benefits in skeletal and striated muscle, neurological disorders and cancer.

Response:
• We thank the reviewer for the meticulous review. We have corrected the spelling errors in the texts.
• We agree with the reviewer’s suggestion on expanding the mechanisms in the cardiovascular system. We decided to divide the cardiovascular section into two main parts: 1) exercise adaptations in increasing cardiac function (line 82-112), and 2) exercise protections in preventing and treating cardiovascular diseases (line 113-180). We also added some specific discussions such as on heart failure and atherosclerosis.

Round 2

Reviewer 1 Report

Thanks for the additions.  Please ensure your headings and subheadings are correct, for instance 3, 4, 6, are main tissues with subheading for each tissue below (e.g. 3.1, 3.2) but then adipose tissue (4.3) is listed under the skeletal muscle section (4)...  Is that correct?  Seems out of place, right?  Moreover, should section 5 be titled "Liver"?  Why is exercise-induced antioxidant actions (7.2) listed under other organs systems (7.0)?

Sorry to say, but the section on skeletal muscle needs more information.  For instance, see your figure, you list contractility for skeletal muscle but don't even mention it in the text.  Moreover, the figure doesn't even mention muscle size...  Muscle contractility and muscle mass are two distant things, and both very important aspects of aging.  Also, considering simply telling the reader to refer to recent reviews for additional information.

Author Response

We thank the reveiwer for the critiques. Here are our point-by-point responses:

Thanks for the additions.  Please ensure your headings and subheadings are correct, for instance 3, 4, 6, are main tissues with subheading for each tissue below (e.g. 3.1, 3.2) but then adipose tissue (4.3) is listed under the skeletal muscle section (4)...  Is that correct?  Seems out of place, right? ==== Heading of the Adipose tissue paragraph was changed to “5” and so as the following paragraphs.

Moreover, should section 5 be titled "Liver"? ====The title of the paragraph on liver was changed to “Liver” (Line 313).

Why is exercise-induced antioxidant actions (7.2) listed under other organs systems (7.0)?====We appreciate this query. We agree with the reviewer. Exercise-induced EcSOD is a rather special mechanism of exercise benefits in which its expression is induced by exercise in the skeletal muscle only but travels through circulation to protect other organs, such as heart and lung. We now put it under the “Skeletal muscle” section.

Sorry to say, but the section on skeletal muscle needs more information.  For instance, see your figure, you list contractility for skeletal muscle but don't even mention it in the text.  Moreover, the figure doesn't even mention muscle size...  Muscle contractility and muscle mass are two distant things, and both very important aspects of aging.  Also, considering simply telling the reader to refer to recent reviews for additional information. ====We thank the reviewer for pointing out the lack of distinguishment in skeletal muscle mass and contractility in the manuscript. We do agree with the reviewer that muscle mass and contractility are two distinct aspects of muscle adaptation to exercise and are both important in healthspan. Therefore, we have revised the skeletal muscle section and reviewed in detail on the two aspects of muscle adaptation (Line 181-258). We also thank the reviewer’s criticism of the figure. Indeed both muscle mass and contractility should be reflected on the figure which we made corresponding changes.